# Vaccine Hesitancy among Medical Students at a Tertiary Hospital—Affiliated Medical School

**DOI:** 10.3390/healthcare11040461

**Published:** 2023-02-05

**Authors:** Ibrahim Omer, Abdullah Alhuzali, Tala Aletani, Zaher Althagafi, Enas Ghulam, Abdullah Awadh

**Affiliations:** 1College of Medicine, King Saud Bin Abdulaziz University for Health Sciences, Riyadh 14611, Saudi Arabia; 2Basic Sciences, College of Science and Health Professions, King Saud Bin Abdulaziz University for Health Sciences, Jeddah 21423, Saudi Arabia; 3King Abdullah International Medical Research Center, Jeddah 21423, Saudi Arabia; 4Basic Medical Sciences, College of Medicine, King Saud Bin Abdulaziz University for Health Sciences, Jeddah 21423, Saudi Arabia

**Keywords:** COVID-19, vaccine hesitancy, medical students, psychological antecedents

## Abstract

Introduction: The coronavirus disease 2019 (COVID-19) caused a global pandemic with long-lasting economic and cultural impacts. International governments have attempted to scale up vaccine production to mitigate this crisis. However, vaccine hesitancy, particularly among healthcare providers, remains an understudied subject that may hinder vaccine effectiveness. Methods: We performed a cross-sectional study to evaluate vaccine hesitancy among medical students using a pre-validated survey based on the 5C model of psychological antecedents, which includes confidence, complacency, constraints, calculation, and collective responsibility. Results: The majority of medical students had high scores for confidence (79.7%), non-complacency (88%), and not having constraints against receiving the COVID-19 vaccine (97.4%). Surprisingly, students had low scores for calculation (38%) and collective responsibility (14.7%). Many predictors of the psychological antecedents included in the 5C model have been reported, including academic year and gender. Conclusion: We observed moderate levels of vaccine hesitancy among the medical students included in our study. We urge medical students to be more aware of community public health concerns. We recommend that authorized institutions lay out urgent reforms to increase awareness of COVID-19 and available vaccines.

## 1. Introduction

Coronavirus disease 2019 (COVID-19) is an acute viral respiratory disease caused by severe acute respiratory syndrome coronavirus-2 (SARS-CoV-2) [1]. In December 2019, China reported the first case of COVID-19 in Wuhan city after an apparent emergence from an animal reservoir. SARS-CoV-2 infections rapidly spread worldwide and were responsible for severe economic and public health burdens [2,3,4]. An alarming rise in infections led to implementing of lockdowns in many nations [5,6,7]. Countries rapidly stockpiled vaccines against COVID-19, which were made mandatory in many parts of the world [8]. The Council on Foreign Relations requested international support for the development of COVID vaccinations, with early vaccine development receiving immense financial support from governments, non-profit organizations, and philanthropic individuals [9].

However, several SARS-CoV-2 variants posing additional threats to public health have since emerged. In December 2021, two novel variants of concern to public health emerged: the delta and omicron variants [10]. The omicron variant is of particular concern due to multiple mutations at the SARS-CoV-2 antigen-binding site, which may reduce vaccine efficacy [11]. Andrews et al. reported lower efficacy of both the BNT162b2 (Pfizer–BioNTech) and ChAdOx1 nCoV-19 (AstraZeneca) against the omicron (B.1.1.529) variant relative to the delta (B.1.617.2) variant [12]. The omicron variant has been reported to have 3–6 times the transmissibility of the delta variant, likely due to the high number of acquired mutations.

Despite the scale of the COVID-19 pandemic, medical professionals have noted vaccine hesitancy among a proportion of individuals. The Strategic Advisory Group of Experts Working Group on Vaccine Hesitancy defines vaccine hesitancy as “the delay in acceptance or refusal of vaccination despite the availability of vaccination services.” Vaccine hesitancy is complex, context-specific, and varies with time, place, and vaccine type. Vaccine hesitancy is influenced by psychological factors, including complacency, convenience, and confidence [13]. Vaccine hesitancy may propagate in specific communities, warranting the Tailoring Immunization Programs to address population disparities in vaccine coverage [14]. The issue of vaccine hesitancy in children first arose with the introduction of vaccination and has been persistently attributed to multiple factors related to parental consent, particularly among under-vaccinated clusters [15]. The clinical importance of vaccine hesitancy is highlighted by a previous study that demonstrated a marked increase in Pertussis infection rates in countries with negative media portrayals of the Pertussis vaccine [16]. Vaccine hesitancy has persisted despite the WHO declaring COVID-19 as a significant threat to global health in 2019 [17]. Vaccine hesitancy is a multifaceted global issue. A recent study observed vaccine hesitancy among 4% of the British population from June 2021 to July 2021, with higher hesitancy rates among minority groups, including Black and Muslim populations, and those of low socioeconomic status [18].

The medical student population, which represents the future frontline of healthcare workers, is an essential population in which to study and evaluate vaccine hesitancy. A study of vaccine hesitancy in medical students in the United States demonstrated that one-fourth were hesitant to receive the COVID-19 vaccine [19]. On the other hand, a separate study found that only 23% of the medical students at an allopathic medical school in Southeast Michigan, U.S.A., would agree to receive COVID-19 vaccination immediately upon approval by the regulatory agencies [20]. In India, however, only one-tenth of medical students reported self-perceived COVID-19 vaccine hesitancy [21]. Hesitancy toward COVID-19 vaccination among medical students appears to vary drastically across regions and nations with no apparent trends.

In cultures of close interdependence between close and extended family members, as in Saudi Arabia, and with faith contributing to the credibility of perspectives from the family members of medical students regarding health issues, families may play essential roles in informing attitudes toward vaccines in the current and future infectious disease pandemics. With an average of 5.7 members per family in Riyadh, the capital of Saudi Arabia [22], the overall effect of family perceptions may delay the achievement of herd immunity, particularly when initiatives aimed at individuals rather than governmental authority are warranted. The trend toward COVID-19 vaccine hesitancy among medical students in Saudi Arabia has yet to be fully characterized. Therefore, using a pre-validated questionnaire, the present study aimed to characterize COVID-19 vaccine hesitancy among medical students at King Saud Bin Abdulaziz University for Health Sciences (KSAU-HS). The present study also aimed to determine the baseline level and causes of COVID-19 vaccine hesitancy among medical students and guide the design of current and future awareness campaigns.

## 2. Materials and Methods

### 2.1. Study Setting and Participants

This cross-sectional study comprised 266 medical students at King Saud Bin Abdulaziz University for Health Sciences (KSAU-HS) College of Medicine in Jeddah and was conducted throughout April 2022. A self-administered electronic questionnaire was used to collect quantitative data hosted through a Google survey webpage^®^. The questions were designed to obtain information regarding vaccine hesitancy among medical students attending KSAU-HS in Jeddah, Saudi Arabia. KSAU-HS was the first health-oriented university founded in the Middle East and houses three campuses in Riyadh, Jeddah, and Al-Ahsa. The university offers several programs for training in healthcare professions, including medicine, dentistry, pharmacy, nursing, and other allied health careers. The present study included students in the first to fifth years at the College of Medicine at KSAU-HS. The total number of medical students attending KSAU-HS during the study period was 843. An online sample size calculator (Raosoft^®^), using a margin of error of 5% and a confidence interval of 95%, was used to determine a recommended minimum sample size of 265.

### 2.2. Data Collection

The study questionnaire consisted of two sections containing 37 items and was adapted from Abd ElHafeez et al. [23] after being granted permission. This is a validated questionnaire that had already been utilized in a previous multinational study conducted in 13 Arab countries [24]. The first section of the questionnaire included 22 items that aimed to collect demographical information from participants, including gender (male/female) and marital status. This section ensured complete confidentiality and participant privacy. The second section of the questionnaire included 15 items in which responders specified their level of agreement with statements related to COVID-19 vaccines using a seven-point Likert scale. Response options were: strongly disagree, disagree, somewhat disagree, neutral, somewhat agree, agree, and strongly agree. Responses were recorded as scores for each item, with the highest score (7 points) allocated to strongly agree and the lowest score to strongly disagree (1 point). The questions belonged to five major domains: vaccine confidence, complacency, constraints, calculation, and collective responsibility (Table 1).

### 2.3. Statistical Analyses

Raw data were examined to identify missing data or inaccuracies prior to statistical analysis. Questionnaire variables were coded to facilitate processing in the statistical computer software, JMP^®^ Version 15 (SAS Institute Inc., Cary, NC, USA). Qualitative variables are presented as frequencies, percentages, or bar graphs. Quantitative variables are presented as the mean with standard deviation (SD). Comparisons and associations between groups were evaluated using the Chi-square and Fisher’s exact tests, respectively. *p*-values less than 0.05 were considered statistically significant. Cutoff values for confidence, complacency, constraints, calculation, and collective responsibility were adopted from Ghazy et al. [26].

### 2.4. Ethical Considerations

The present study was approved by the Institutional Review Board of King Abdullah International Medical Research Center (No JED-22-427780-37278). The work described here was carried out in accordance with The Code of Ethics of the World Medication Association (Declaration of Helsinki).

## 3. Results

### 3.1. Participant Characteristics

A total of 266 medical students were included in the present study, with a response rate of 100%. The majority of participants were male (54.9%), in the third year (35.7%), single (98.9%), knew a family member who was infected with SARS-CoV-2 before (75.2%), or after receiving the vaccine (84.2%) and were aware that multiple COVID-19 vaccines existed (99.3%). The mean age of participants was 21.3 (±1.7) years. All participants had been vaccinated against COVID-19 (100%). The majority of participants reported a preference toward the Pfizer-BioNTech vaccine (87.2%), had searched for data pertaining to the vaccine before receiving the vaccine (64.7%), and described the provision of the COVID-19 vaccine free of cost as an influential factor in receiving a vaccination (63.2%). Only a minority of participants had a chronic illness (9.4%), knew a family member or a friend who passed away due to COVID-19 (28.6%), or had read vaccine precautions before receiving the vaccine (39.9%; Table 2).

### 3.2. COVID-19 Psychological Antecedents

The majority of the participants in the present study were confident (79.7%), non-complacent (88%), and did not have constraints against receiving the COVID-19 vaccine (97.4%). Participants had low scores for calculation (38%) and collective responsibility (14.7%) (Figure 1).

### 3.3. Bivariate Analyses

Factors significantly associated with confidence included academic year (*p* < 0.001), prior SARS-CoV-2 infection (*p* = 0.021), and the provision of a free-of-cost COVID-19 vaccine as an influential factor in receiving vaccination (*p* = 0.025; Table 3). Factors significantly associated with complacency included prior family SARS-CoV-2 infection (*p* = 0.001) and a family member or friend passing away due to COVID-19 (*p* = 0.047; Table 4). Factors significantly associated with constraints included gender (*p* = 0.003), prior SARS-CoV-2 infection (*p* < 0.001), family member or friend passing away due to COVID-19 (*p* = 0.003), reading precautions prior to receiving the COVID-19 vaccine (*p* = 0.012), and searching for information related to COVID-19 vaccine (*p* = 0.048; Table 5). Factors significantly associated with calculation included prior SARS-CoV-2 infection (*p* = 0.022), prior family SARS-CoV-2 infection (*p* = 0.029), perceived best COVID-19 vaccine (*p* = 0.004), reading precautions prior to receiving the COVID-19 vaccine (*p* < 0.001), and searching for information related to COVID-19 vaccine (*p* < 0.001; Table 6). Only free COVID-19 vaccine being an influential factor in receiving it (*p* = 0.022) was reported as a significantly factor associated with collective responsibility (Table 7).

### 3.4. Multivariate Analysis

Predictors of confidence included being in the second (OR, 7.042; 95% CI, 1.55–32.04) or third year (OR, 3.349; 95% CI, 1.51–7.45) of medical school, knowledge of previous confirmed SARS-CoV-2 infection (OR, 2.49; 95% CI, 0.66–9.44), no known previous SARS-CoV-2 infection (OR, 4.37; 95% CI, 1.17–16.31), and free COVID-19 vaccine as an influential factor in receiving it (OR, 2.00; 95% CI, 1.04–3.86; Table 8). 

Predictors of complacency included age, knowledge of previous confirmed SARS-CoV-2 infection in a family member (OR, 0.07; 95% CI, 0.01–0.36), no known previous SARS-CoV-2 infection in a family member (OR, 0.08; 95% CI, 0.01–0.50), and perceived bias toward Pfizer-BioNTech© as the best available vaccine (OR, 0.06; 95% CI: 0.01–0.34; Table 8).

Predictors of calculation included confirmed prior SARS-CoV-2 infection (OR, 0.23; 95% CI, 0.07–0.83), having read vaccine precautions prior to receiving the vaccine (OR, 1.80; 95% CI, 1.03–3.15), and having searched for information related to COVID-19 vaccination online (OR, 2.65; 95% CI, 1.42–4.95; Table 8).

No significant predictors of constraint or collective responsibility related to the COVID-19 vaccine were identified (Table 8).

## 4. Discussion

During the early days of the COVID-19 pandemic, there was great interest in quantifying the hesitancy of the general public toward receiving the COVID-19 vaccine. The risks associated with vaccine administration have been identified as factors contributing to vaccine hesitancy [27] in addition to other factors, such as low trust in science, accessibility concerns, prior experiences, and historical influences [28]. In a survey of approximately 20,000 adults conducted in July 2020 in 27 countries, approximately 74% of respondents intended to accept the COVID-19 vaccine [29]. In a systematic review, the highest rates of vaccine acceptance were reportedly observed in China, Ecuador, Indonesia, and Malaysia, all of which had >90% vaccine acceptance, whereas nations such as the U.S.A., Jordan, Poland, and Italy ranked the lowest in vaccine acceptance [30,31,32,33]. However, as the pandemic progressed, studies showed that a substantial number of healthcare workers were vaccine-hesitant, with fears regarding the side effects of vaccines found to be most prominent [34]. Vaccine hesitancy was also observed among medical students, with nearly one-quarter of medical school students in the U.S. hesitant to receive the COVID-19 vaccine even after the Food and Drug Administration (FDA) approval [20].

Healthcare professionals have increased exposure to at-risk patient populations. Despite this, a substantial proportion of healthcare professionals are vaccine-hesitant [35]. Several reports, such as the December 2020 Kaiser Family Foundation’s Poll, have demonstrated that vaccine hesitancy among healthcare workers is similar to that of the general population, with 29% of healthcare workers vaccine-hesitant compared to only 27% of the general population [36]. Similarly, approximately 77.8% of residents and 37.5% of staff members across multiple nursing facilities in the U.S. received at least one dose of the COVID-19 vaccine (a total of 76,741 participants) [31]. As healthcare workers are a trusted source of information for patients and their social circles, healthcare workers’ views are relevant to any vaccination campaign in the present or future pandemic responses. Thus, understanding the motives underlying hesitancy to vaccines is essential for guiding corrective public health policies and practices.

The present study is unique because it used the 5C scale to assess psychological antecedents regarding COVID-19 vaccination in medical students attending a tertiary hospital-affiliated medical school in Saudi Arabia. This scale was able to carefully and specifically quantify attitudes toward vaccination uptake among our target population and provided a comparable, quantifiable set of characteristics that apply to future studies in similar target populations. Differences between the global non-healthcare population and healthcare workers may be attributed to geographical differences, as geographic location appears to correlate more strongly with vaccine acceptance [29]. We attribute the disparity in confidence scores to medical students being able to trust medical institutions and better understand the implications of infectious diseases as they progress through their education from the early years of medical school to later ones.

Medical students with previous SARS-CoV-2 infections had high vaccine confidence as they were more aware of the potential severity of COVID-19 and its implications for the community. Additionally, most medical students with previous SARS-CoV-2 infection had no constraints to receiving the vaccine, most likely due to firsthand experience of the symptoms of COVID-19 and understanding the urgency of vaccination against COVID-19. Those previously infected were also more likely to recognize the significance of the government’s efforts to vaccinate against COVID-19 and the government-provided services during the pandemic. Elharake et al. [37] reported that trust in the authorities was a major reason for health workers in Saudi Arabia having a higher acceptance rate for the COVID-19 vaccine, a finding supported by the results of the present study.

Most students with family members who had been infected with SARS-CoV-2 or had a family member or friend pass away due to COVID-19 had no complacency toward the COVID-19 vaccine. Complacent individuals often believe that the vaccine is unnecessary and that their immune system can protect them from the disease. However, as the respondents were medical students who studied immunology and were personally exposed to cases of COVID-19, they were less skeptical regarding the importance of the COVID-19 vaccine. Further, a previous study conducted in Arab countries found that working in healthcare was associated with vaccine acceptance [38]. Additionally, students are more likely to educate themselves regarding the importance and efficiency of COVID-19 vaccines as they are more empathetic toward the cause. Furthermore, students who read COVID-19 vaccine precautions provided by authorities and searched for information on COVID-19 were more likely to calculate the risks and benefits of vaccination and have fewer constraints.

In Saudi Arabia, COVID-19 vaccines were provided free of charge to all citizens and residents. Thus, students were obligated to receive the COVID-19 vaccine and were more confident in receiving it as it was free of charge. Furthermore, a multinational study conducted in 13 Arab countries found that vaccine confidence was highest in the United Arab Emirates, followed by Saudi Arabia [24]. This finding indicates that the Saudi Arabian population has a high acceptance of the COVID-19 vaccine.

The survey used in the present study included four vaccines as options: the Pfizer-BioNTech, Oxford-AstraZeneca, Moderna, and Sinophram vaccines. However, most respondents had only received the Pfizer-BioNTech or Oxford-AstraZeneca vaccines or both, as the Saudi Arabian government initially only provided these two vaccines and only provided the Moderna vaccine in the late stages of the pandemic. Students were more inclined to choose the Pfizer-BioNTech or Oxford-AstraZeneca vaccines due to excessive reporting regarding the Pfizer-BioNTech and Oxford-AstraZeneca vaccines in the media and the experiences of family members and friends after receiving these vaccines. These factors made participants less complacent about these two vaccines and promoted the development of a positive attitude toward vaccination against COVID-19.

Further, there was a risk of selection bias as the survey was predominantly distributed through social media platforms and e-mail. As a self-reported questionnaire was used to collect data, the findings of the present study may have been biased by social desirability. However, the findings were consistent with previous studies that reported the behavioral factors associated with COVID-19 vaccination. It is of note as well that this survey was administered during the COVID-19 pandemic, which might have affected the knowledge and attitudes of our participants toward the survey.

## 5. Conclusions

The results of the present study demonstrate moderate levels of vaccine hesitancy among medical students in Saudi Arabia, with low levels of collective responsibility and calculation related to the COVID-19 vaccine. We look forward to institutional policies that will reduce vaccine hesitancy among medical students based on the predictors of hesitancy identified in the present study.

## Figures and Tables

**Figure 1 healthcare-11-00461-f001:**
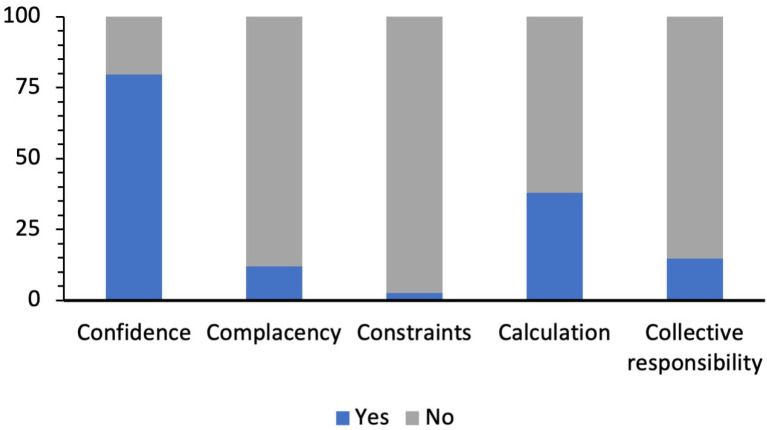
Overall results for Coronavirus disease 2019 psychological antecedents.

**Table 1 healthcare-11-00461-t001:** 5C Psychological Antecedents Definitions.

Term	Definition
Confidence	“Trust in (i) the effectiveness and safety of vaccines, (ii) the system that delivers them, including the reliability and competence of the health services and health professionals, and (iii) the motivations of policy-makers who decide on the need of vaccines.” [25]
Complacency	“Complacency exists where perceived risks of vaccine-preventable diseases are low and vaccination is not deemed a necessary preventive action.” [25]
Constraints	“Constraints can manifest in limited physical availability, affordability and willingness-to-pay, geographical accessibility, ability to understand (language and health literacy) and appeal of immunization service.” [25]
Calculation	“Individuals’ engagement in extensive information searching.” [26]
Collective responsibility	“Willingness to protect others by one’s own vaccination by means of herd immunity.” [26]

**Table 2 healthcare-11-00461-t002:** Participant characteristics.

Variable		n	%
		(n = 266)	
Sex	Female	120	45.1
	Male	146	54.9
Academic year	1st	82	30.8
	2nd	33	12.4
	3rd	95	35.7
	4th	27	10.2
	5th	29	10.9
Age (mean ± SD)	21.3 ± 1.7		
Social status	Single	263	98.9
	Married	3	1.1
Chronic illness(es)		25	9.4
Annual receipt of the influenza vaccine		97	36.5
Previous infection with SARS-CoV-2 ^1^		89	33.5
Family member with previous SARS-CoV-2 infection		200	75.2
Family member or friend passed away due to COVID-19 ^2^		76	28.6
Aware of the availability of multiple COVID-19 vaccines		264	99.3
Vaccinated against COVID-19		266	100
Perceived best vaccine			
Pfizer-BioNTech Oxford-AstraZeneca Moderna Sinopharm		232	87.2
		23	8.7
		7	2.6
		4	1.5
Knew of a family member or friend who had been infected with COVID-19 after receiving vaccination		224	84.2
Read precautions provided by the local authorities related to COVID-19 vaccination		106	39.9
Searched for information related to COVID-19 vaccination online		172	64.7
Free COVID-19 vaccine as an influential factor in receiving it		168	63.2

^1^ SARS-CoV-2, severe acute respiratory syndrome coronavirus; ^2^ COVID-19, coronavirus disease 2019.

**Table 3 healthcare-11-00461-t003:** Subgroup analysis–confidence.

Variable	Confidence		
	n (%)		*p*-Value
	Yes	No	
Sex			
Female	92 (76.7)	28 (23.3)	0.265
Male	120 (82.2)	26 (17.8)	
Academic Year			
2nd year	54 (65.9)	28 (34.2)	<0.001 ^1^
3rd year	31 (93.9)	2 (6.1)	
4th year	84 (88.4)	11 (11.6)	
5th year	22 (81.5)	5 (18.5)	
6th year	21 (72.4)	8 (27.6)	
Chronic illness(es)	20 (80.0)	5 (20.0)	0.969
Annual receipt of influenza vaccine	79 (81.4)	18 (18.6)	0.592
Previous infection with SARS-CoV-2 *	65 (73.0)	24 (27.0)	0.021 ^1^
Family member with previous SARS-CoV-2 infection	154 (77.0)	46 (23.0)	0.161
Family member or friend passed away due to COVID-19 **	58 (76.3)	18 (23.7)	0.057
Perceived Best COVID-19 Vaccine			
Pfizer-BioNTech	187 (80.6)	45 (19.4)	0.202 ^2^
Oxford-AstraZeneca	19 (82.6)	4 (17.4)	
Moderna	4 (57.1)	3 (42.9)	
Sinopharm	2 (50.0)	2 (50.0)	
Knew a family member or friend who had been infected with SARS-CoV-2 after receiving vaccination	174 (77.7)	50 (22.3)	0.059 ^2^
Read precautions provided by the local authorities for COVID-19 vaccination	88 (83.0)	18 (17.0)	0.273
Searched for information related to COVID-19 vaccination online	140 (81.4)	32 (18.6)	0.352
Free COVID-19 vaccine as an influential factor in receiving it	141 (83.9)	27 (16.1)	0.025 ^1^

^1^ Statistically significant; ^2^ Fisher’s exact test; * SARS-CoV-2, severe acute respiratory syndrome coronavirus 2; ** COVID-19, coronavirus disease 2019.

**Table 4 healthcare-11-00461-t004:** Subgroup analysis–complacency.

Variable	Complacency		
	n (%)		*p*-Value
	Yes	No	
Sex			
Female	16 (13.3)	104 (86.7)	0.554
Male	16 (11)	130 (89)	
Academic Year			
2nd year	11 (13.4)	71 (86.6)	0.919 ^2^
3rd year	5 (15.2)	28 (84.8)	
4th year	11 (11.6)	84 (88.4)	
5th year	2 (7.41)	25 (92.6)	
6th year	3 (10.4)	26 (89.7)	
Chronic illness(es)	4 (16.0)	21 (84.0)	0.518 ^2^
Annual receipt of influenza vaccine	9 (9.3)	88 (90.7)	0.296
Previous infection with SARS-CoV-2 *	14 (15.7)	75 (84.3)	0.404
Family member with previous SARS-CoV-2 * infection	22 (11.0)	178 (89.0)	0.001 ^1^
Family member or friend passed away due to COVID-19 **	11 (14.5)	65 (85.5)	0.047 ^1^
Perceived Best COVID-19 Vaccine			
Pfizer-BioNTech	25 (10.8)	207 (89.2)	0.052 ^2^
Oxford-AstraZeneca	3 (13.0)	20 (87.0)	
Moderna	3 (42.9)	4 (57.1)	
Sinopharm	1 (25.0)	3 (75.0)	
Knew a family member or friend who had been infected with SARS-CoV-2 * after receiving vaccination	30 (13.4)	194 (86.6)	0.192 ^2^
Read precautions provided by the local authorities for COVID-19 ** vaccination	14 (13.2)	92 (86.8)	0.631
Searched for information related to COVID-19 ** vaccination online	21 (12.2)	151 (87.8)	0.903
Free COVID-19 ** vaccine as an influential factor in receiving it	19 (11.3)	149 (88.7)	0.636

^1^ Statistically significant; ^2^ Fisher’s exact test; * SARS-CoV-2, severe acute respiratory syndrome coronavirus 2; ** COVID-19, coronavirus disease 2019.

**Table 5 healthcare-11-00461-t005:** Subgroup analysis—constraints.

Variable	Constraints		
	n (%)		*p*-Value
	Yes	No	
Sex			
Female	7 (5.8)	113 (94.2)	0.003 ^1,2^
Male	0 (0)	146 (100.0)	
Academic Year			
2nd year	2 (2.4)	80 (97.6)	0.821 ^2^
3rd year	1 (3.0)	32 (97.0)	
4th year	4 (4.2)	91 (95.8)	
5th year	0 (0.0)	27 (100.0)	
6th year	0 (0.0)	29 (100.0)	
Chronic illness(es)	1 (4.0)	24 (96.0)	0.503 ^2^
Annual receipt of influenza vaccine	3 (3.1)	94 (96.9)	0.708 ^2^
Previous infection with SARS-CoV-2 *	7 (7.9)	82 (92.1)	0.001 ^1,2^
Family member with previous SARS-CoV-2 * infection	7 (3.5)	193 (96.5)	0.467 ^2^
Family member or friend passed away due to COVID-19 **	6 (7.9)	70 (92.1)	0.007 ^1,2^
Perceived Best COVID-19 Vaccine			
Pfizer-BioNTech	6 (2.6)	226 (97.4)	0.621 ^2^
Oxford-AstraZeneca	1 (4.4)	22 (95.6)	
Moderna	0 (0.0)	7 (100.0)	
Sinopharm	0 (0.0)	4 (100.0)	
Knew a family member or friend who had been infected with SARS-CoV-2 *after receiving vaccination	7 (3.1)	217 (96.9)	0.601 ^2^
Read precautions provided by the local authorities for COVID-19 ** vaccination	6 (5.7)	100 (94.3)	0.017 ^1,2^
Searched for information related to COVID-19 ** vaccination online	7 (4.1)	165 (95.9)	0.053 ^2^
Free COVID-19 ** vaccine as an influential factor in receiving it	5 (3.0)	163 (97.0)	1.00 ^2^

^1^ Statistically significant; ^2^ Fisher’s exact test; * SARS-CoV-2, severe acute respiratory syndrome coronavirus 2; ** COVID-19, coronavirus disease 2019.

**Table 6 healthcare-11-00461-t006:** Subgroup analysis–calculation.

Variable	Calculation		
	n (%)		*p*-Value
	Yes	No	
Sex			
Female	43 (35.8)	77 (64.2)	0.515
Male	58 (39.7)	88 (60.3)	
Academic Year			
2nd year	29 (35.4)	53 (64.6)	0.280
3rd year	8 (24.2)	25 (75.8)	
4th year	43 (45.3)	52 (54.7)	
5th year	10 (37.0)	17 (63.0)	
6th year	11 (37.9)	18 (62.1)	
Chronic illness(es)	8 (32.0)	17 (68.0)	0.518
Annual receipt of influenza vaccine	35 (36.1)	62 (63.9)	0.631
Previous infection with SARS-CoV-2 *	25 (28.1)	64 (71.9)	0.022 ^1^
Family member with previous SARS-CoV-2 * infection	67 (33.5)	133 (66.5)	0.024 ^1,2^
Family member or friend passed away due to COVID-19 **	34 (44.7)	42 (55.3)	0.294 ^2^
Perceived Best COVID-19 Vaccine			
Pfizer-BioNTech	82 (35.3)	150 (64.7)	0.003 ^1,2^
Oxford-AstraZeneca	13 (56.5)	10 (43.5)	
Moderna	6 (85.7)	1 (14.3)	
Sinopharm	0 (0.0)	4 (100.0)	
Knew a family member or friend who had been infected with SARS-CoV-2 *after receiving vaccination	82 (36.6)	142 (63.4)	0.290
Read precautions provided by the local authorities for COVID-19 ** vaccination	54 (50.9)	52 (49.1)	<0.001 ^1^
Searched for information related to COVID-19 ** vaccination online	80 (46.5)	92 (53.5)	<0.001 ^1^
Free COVID-19 ** vaccine as an influential factor in receiving it	58 (34.5)	110 (65.5)	0.129

^1^ Statistically significant; ^2^ Fisher’s exact test; * SARS-CoV-2, severe acute respiratory syndrome coronavirus 2; ** COVID-19, coronavirus disease 2019.

**Table 7 healthcare-11-00461-t007:** Subgroup analysis—collective responsibility.

Variable	Collective Responsibility		
	n (%)		*p*-Value
	Yes	No	
Sex			
Female	12 (10)	108 (90)	0.051
Male	27 (18.5)	119 (81.5)	
Academic Year			
2nd year	15 (18.3)	67 (81.7)	0.683 ^2^
3rd year	5 (15.2)	28 (84.8)	
4th year	14 (14.7)	81 (85.3)	
5th year	3 (11.1)	24 (88.9)	
6th year	2 (6.9)	27 (93.1)	
Chronic illness(es)	3 (12.0)	22 (88.0)	0.693
Annual receipt of influenza vaccine	10 (10.3)	87 (89.7)	0.128
Previous infection with SARS-CoV-2 *	14 (15.7)	75 (84.3)	0.199
Family member with previous SARS-CoV-2 infection	30 (15.0)	170 (85.0)	0.480
Family member or friend passed away due to COVID-19 **	16 (21.1)	60 (78.9)	0.108
Perceived Best COVID-19 Vaccine			
Pfizer-BioNTech	34 (14.7)	198 (85.3)	0.601 ^2^
Oxford-AstraZeneca	3 (13.0)	20 (87.0)	
Moderna	2 (28.6)	5 (71.4)	
Sinopharm	0 (0.0)	4 (100.0)	
Knew a family member or friend who had been infected with SARS-CoV-2 after receiving vaccination	32 (14.3)	192 (85.7)	0.689
Read precautions provided by the local authorities for COVID-19 vaccination	16 (15.1)	90 (84.9)	0.871
Searched for information related to COVID-19 vaccination online	25 (14.5)	147 (85.5)	0.937
Free COVID-19 vaccine as an influential factor in receiving it	31 (18.5)	137 (81.5)	0.022 ^1^

^1^ Statistically significant; ^2^ Fisher’s exact test; * SARS-CoV-2, severe acute respiratory syndrome coronavirus 2; ** COVID-19, coronavirus disease 2019.

**Table 8 healthcare-11-00461-t008:** Logistic regression analysis.

Independent Variable		Odds Ratio	*p*-Value	95% CI for Odds Ratio	Goodness-of-Fit
Confidence					R^2^ = 0.108 Entropy R^2^ = 0.108 Generalized R^2^ = 0.162
Academic year	2nd year (a)	7.042	0.012 ^1^	(1.548, 32.044)	
	3rd year (a)	3.349	0.003 ^1^	(1.507, 7.445)	
	4th year (a)	1.803	0.295	(0.598, 5.433)	
	5th year (a)	1.222	0.683	(0.465, 3.201)	
Previous infection with SARS-CoV-2 *	No (b)	4.368	0.028 ^1^	(1.169, 16.314)	
	Yes (b)	2.487	0.180	(0.655, 9.444)	
Free COVID-19 ** vaccine as an influential factor in receiving it	Yes (c)	2.002	0.038 ^1^	(1.038, 3.860)	
Complacency					
Age		0.689	0.017 ^1^	(0.509, 0.934)	R^2^ = 0.108 Entropy R^2^ = 0.108 Generalized R^2^ = 0.147
Family member previously infected with SARS-CoV-2	No (b)	0.080	0.007 ^1^	(0.012, 0.496)	
	Yes (b)	0.066	0.002 ^1^	(0.012, 0.357)	
Perceived best COVID-19 vaccine	Sinopharm (d)	0.195	0.269	(0.195, 42.853)	
	Pfizer-BioNTech (d)	0.057	0.002 ^1^	(0.009, 0.342)	
	Oxford-AstraZeneca (d)	0.346	0.441	(0.023, 5.134)	
Calculation					R^2^ = 0.084 Entropy R^2^ = 0.084 Generalized R^2^ = 0.143
Previous infection with SARS-CoV-2	No (b)	0.486	0.245	(0.144, 1.641)	
	Yes (b)	0.232	0.024 ^1^	(0.065, 0.826)	
Read precautions provided by the local authorities for COVID-19 vaccination	Yes (c)	1.798	0.039 ^1^	(1.028, 3.148)	
Searched for information related to COVID-19 vaccination online	Yes (c)	2.651	0.002 ^1^	(1.418, 4.953)	

Lowercase letters indicate comparisons with corresponding group. a, 1st year; b, I don’t know; c, no; d, Moderna; ^1^ Statistically significant; * SARS-CoV-2, severe acute respiratory syndrome coronavirus 2; ** COVID-19, coronavirus disease 2019.

## Data Availability

All cleaned data have been included in this manuscript. All raw data will be available upon request.

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
