# Peer review of "Vaccine Hesitancy among Medical Students at a Tertiary Hospital—Affiliated Medical School"

_healthcare, 2023, doi:10.3390/healthcare11040461_

Round 1

Reviewer 1 Report

The article concerns the interesting topic of study on "Vaccine Hesitancy among Medical Students at a Tertiary Hospital-Affiliated Medical School." The subject of the article is current and in line with the thematic scope of the journal. For the paper to reach the desired quality, the authors should handle the following issues well.

The last sentence of the summary part of the study can be reviewed and concluded with statements such as 'we recommend to authorized institutions.

To understand the effects/relationships of the variables with each other, correlation values can be calculated by Pearson or spearman rho tests.

The R2, R2 (adjusted), and R2 (predicted) values should be shared to validate the validity of the statistical analysis performed in this study.

Is there any outlier data in the study data? The proportions of these data, if any, should be shared.

Are there any limits to the work? If there is, it should be shared. For example, the effect of the corona is observed in the period when the data is collected. Does it mean statistically significant between the period when the corona epidemic was intense and when the data were collected?

The data included 266 medical students at the King Saud Bin Abdulaziz Health Sciences University (KSAU-HS) School of Medicine in Jeddah and was conducted through April 2022. The authors collected data of the study in just one month. Supporting statements are needed to measure the statistical reliability of this study, which was completed in a short time and with little data.

Some grammatical/typing errors should be corrected.

The authors well handled the conclusion and discussion part of the study. However, to confirm the result of this study, a comparison should be made according to the effects of similar analyses.

Reviewer 2 Report

Congratulations on the submission of the article titled ‘Vaccine Hesitancy among Medical Students at a Tertiary Hospital-Affiliated Medical School

The study is interesting and relevant. However, there are major revisions needed. These are listed below.

Title of article:

I do not think this title is appropriate for the article. Initially, the title looks eye-catching, but on reading through the article and realising that 100% of the study population had been vaccinated, is the title still appropriate?

Line 294 in the discussion suggests that the students were encouraged, not mandated to receive the vaccine and they all received it. So can you really say there was vaccine hesitancy in the study population as reflected in the title?

Abstract

No issues were identified. However, the word ‘herein’ in line 17 can be removed as the sentence will still be complete without it.

Introduction

Lines 47-49: The statement is unclear. Did the referenced study report lower efficacy of both vaccines against the 2 variants? This should be made clearer in the text.

Materials and Methods

The selection method should be further clarified. There were 843 eligible participants, with a minimum size of 265. Was the questionnaire sent to all 843 students or only 266? This information may have an impact on the response rate which has been reported as 100% (line 144).

Line 112: This suggested that there were 36 items on the questionnaire, line 115 suggested that the first section included 22 items and line 118 suggests that the second section included 15 items. 22+15 is 37, not 36. This needs to be clarified.

It would be very helpful for the questionnaire used in the study to be presented to the readers as supplementary material.

Line 114: ‘survey’ can be changed to ‘questionnaire’

Results

Line 153: The word ‘However’ can be removed from the start of the sentence.

Table 2: males were 146, not 246.

Discussion

Line 227: In the systemic review quoted, it would have been more relevant to the discussion if the reasons for vaccine hesitancy were stated, if known.

The paragraph starting from line 258: The reason given for the difference between males and females receiving vaccination, while relevant for the general Saudi population is unlikely not correct for the population studied since 100% of them had received the vaccine. Secondly, this is a population of medical students and it is most likely that they will have easy access to health facilities by virtue of their course of study.

Conclusion

You have concluded that there was moderate level of vaccine hesitancy, how was this conclusion arrived at, as all these students had received the vaccine?

References

There are errors in the references and all the references should be checked again for correctness. Some of the errors are listed below.

Line 49: Andrews et al are quoted as reference 12 in the text, but it is reference 13 on the list of references.

Line 58: The paper that is quoted is reference 14, not 13.

Line 65: The reference to pertussis vaccination is reference 19 on the list of references, not 16.

Round 2

Reviewer 2 Report

The suggested corrections have been carried out. No further comments from me.